# Fermentation of Soybean Meal with *Lactobacillus acidophilus* Allows Greater Inclusion of Vegetable Protein in the Diet and Can Reduce *Vibrionacea* in the Intestine of the South American Catfish (*Rhamdia quelen*)

**DOI:** 10.3390/ani12060690

**Published:** 2022-03-09

**Authors:** Nandara Soares de Oliveira, Natalia Ha, Larissa da Cunha, Luiz Augusto Cipriani, André Thaler Neto, Everton Skoronski, Enric Gisbert, Thiago El Hadi Perez Fabregat

**Affiliations:** 1Departamento de Produção Animal, Centro de Ciências Agroveterinárias, Universidade do Estado de Santa Catarina-UDESC, Avenue Luiz de Camões, 2090, Lages 88520-000, SC, Brazil; nandara.soares@hotmail.com (N.S.d.O.); ha.natalia@yahoo.com.br (N.H.); laridacunha@gmail.com (L.d.C.); ciprianiaugustoluiz@gmail.com (L.A.C.); andre.thaler@udesc.br (A.T.N.); everton.skoronski@udesc.br (E.S.); 2IRTA, Centre de Sant Carles de la Ràpita, Aquaculture Program, Carretera Poble Nou, km 5.5, 43540 Sant Carles de la Ràpita, Spain; enric.gisbert@irta.cat

**Keywords:** digestive enzymes, fermented soybean, fish nutrition, intestinal histomorphometry

## Abstract

**Simple Summary:**

There is a demand to replace fishmeal with protein sources of plant origin in fish feeds. Biotechnology strategies, such as fermentation, can improve the bioavailability of plant proteins and decrease the presence of antinutrients, optimizing the results obtained. Fermented soybean meal has already been evaluated for different fish species as a replacement for fishmeal, and there is evidence that it can improve the intestinal health of animals. *Lactobacillus acidophilus* is a strain used as a probiotic in fish feeding but it remains to be evaluated as a potential fermentation bacterium for feed ingredients. This study aimed to evaluate the effect of diets containing different inclusion levels (0%, 7%, 14%, 21% and 28%) of soybean meal fermented by *L. acidophilus* (SMFL) on the zootechnical performance and intestinal health of South American catfish juveniles (*Rhamdia quelen*). The inclusion of SMFL up to 21% in replacement of fish meal did not affect the zootechnical performance of fish and also decreased the concentration of *Vibrionaceae* bacteria present in the intestine compared to the control group. The results demonstrate that fermentation with *L. acidophilus* enables greater inclusion of soybean protein in South American catfish diets and promotes the control of intestinal pathogenic bacteria.

**Abstract:**

The objective of this study was to evaluate the effect of diets containing different inclusion levels (0%, 7%, 14%, 21% and 28%) of soybean meal fermented by *Lactobacillus acidophilus* (SMFL) on the zootechnical performance and intestinal health of South American catfish juveniles (*Rhamdia quelen*). The experimental design was completely randomized with five treatments and four replications and lasted 56 days. Five isoproteic (39% crude protein) and isoenergetic (4300 kcal of gross energy kg^−1^) diets were formulated where SMFL was included in replacement of fish meal. Two hundred forty South American catfish juveniles (3.0 ± 0.5 g) were distributed in 20 tanks (70 L) connected in a recirculation aquaculture system. At the end of the experiment, the inclusion of SMFL up to 21% in replacement of fish meal did not affect the zootechnical performance and also decreased the concentration of *Vibrionaceae* bacteria present in the intestine compared to the control group. The amount of total lactic and heterotrophic bacteria, the enzymatic activity and the intestinal morphometry did not differ between dietary treatments. The results demonstrate that fermentation with *Lactobacillus acidophilus* enables greater inclusion of soybean protein in South American catfish diets and promotes the control of intestinal pathogenic bacteria.

## 1. Introduction

Soybean meal is an ingredient that is the most common vegetal protein source in animal nutrition due to its availability and balanced amino acid profile [1,2]. However, the palatability of soybean meal is low [3], and there are problems with antinutritional factors that limit its use in feeding aquatic organisms [4]. Some of the antinutritional factors, such as trypsin inhibitors and hemagglutinants, are inactivated by the toasting process [5]. However, other factors are heat-stable, such as saponins, non-starch polysaccharides, antigenic proteins, and some phenolic compounds. Furthermore, the excess of oligosaccharides can impair the digestive process [5], and the presence of antinutritional factors such as phytic acid can impair protein digestibility [1].

Fermentation is a method that can enhance the quality of feed ingredients, inactivate antinutritional factors, improve nutritional bioavailability and increase soluble proteins and small peptides [6]. Fermented products are foods or beverages that serve as substrates for microorganisms, which through enzymes (amylases, proteases and lipases, among others), transform and reduce the size of organic compounds [7]. Fermented foods are part of the human diet due to their nutritional and functional qualities [8]. In recent years, the use of fermented ingredients as functional diets in aquaculture species has attracted the attention of academia and industry [9].

Microbial fermentation is carried out using fungi or bacterial strains [8]. *Lactobacillus* spp. are widely used in the process as they are considered nonpathogenic and safe bacteria [10]. Like other lactic acid bacteria, *Lactobacillus* spp. can inhibit other competing microorganisms by combining rapid carbohydrate utilization and lactic acid production [11]. Several strains of *Lactobacillus* spp. have functional properties and can be used in fermentation processes. *Lactobacillus acidophilus* are important microorganisms in the composition of the human intestinal microbiota [12]. This strain is used as a probiotic in fish feeding [10] as it has therapeutic activities on intestinal health and produces a variety of antimicrobial compounds that are effective against pathogenic bacteria [13]. Positive results have already been obtained with *Lactobacillus* spp. mixed cultures in the fermentation of soybean meal [14,15], but the use of *L. acidophilus* remains to be evaluated as a potential fermentation bacterium for feed ingredients.

Fermentation is an advantageous technique for nutritional improvement of soybean meal, through biodegradation and reduction of oligosaccharides and phytic acid, and for enhancement of the amino acid profile [16,17,18,19]. Fermentation also causes a release of bioactive peptides, which have functional properties in animal metabolism by modulating the immune system [20]. In studies with fish, the use of fermented soybean meal showed positive results on weight gain [21,22,23,24]. Enhancement in the intestinal health of fish fed with fermented soybean was observed, evidenced by the increase in beneficial bacteria in the intestinal microbiota [23,25,26], by the increase in the activity of gastrointestinal enzymes [27,28,29,30] and by the improvement of intestinal condition histological morphometric indexes [27,31,32,33,34].

The South American catfish (*Rhamdia quelen*) is an omnivorous species with a tendency to exhibit carnivore behavior [35]. Regarding nutrition, the South American catfish is considered a demanding fish in terms of quality [36] and quantity of protein [37]. Diets for this species are generally formulated with high levels of animal protein sources, whereas when fishmeal has been replaced with alternative protein sources such as soybean meal, the zootechnical performance of this species has been compromised [37,38]. In order to overcome such a limitation, fermentation may be used as a strategy for enabling higher levels of soy protein in compound feeds and for benefiting the intestinal health of the South American catfish. Therefore, the objective of this work is to evaluate the effect of diets containing different levels of inclusion of soybean meal fermented by *L. acidophilus* (SMFL) on the zootechnical performance and intestinal health condition of South American catfish juveniles.

## 2. Materials and Methods

### 2.1. Experimental Design

The experiment was carried out at the Santa Catarina State University (UDESC) Fish Farming Laboratory, in Lages, SC, Brazil. The study was approved by the Ethics Committee on Animal Use (CEUA) of UDESC (protocol number 4384210621). SMFL was evaluated at four levels (7%, 14%, 21% and 28%) of inclusion compared to a control diet (0%) during 56 days. The experimental design was completely randomized, with five treatments and four replications.

### 2.2. Production and Characterization of Fermented Soybean Meal

Soybean meal was fermented using an adaptation of the methodology of Azarm and Lee [39]. In brief, soybean meal and bacteria were purchased from local suppliers. Initially, the soybean meal was autoclaved (AV-100, Tecnal, Piracicaba, Brazil) at 100 °C for 20 min. For this purpose, the soybean meal was autoclaved at a temperature of 100 °C for 20 min and inoculated with *L. acidophilus* (strain *LA14*-1.10^9^ CFU g^−1^, Aché, São Paulo, Brazil) in the proportion of 44 g kg^−1^. Then, deionized water was added in the proportion of 50% of the soybean meal weight (model ML 600, Marte, Brazil), and the mixture was placed in trays previously sterilized with 70% ethanol. The mixture was distributed in trays in order to obtain an average thickness of two centimeters and then placed in an incubator (SSDcr-336L, SolidSteel, Piracicaba, Brazil) at 36 °C. Every 12 h, the material was weighed to quantify the evaporated water, replaced and homogenized to maintain moisture at 50%. Every 24 h, up to a maximum time of 96 h, samples were collected from the material that was dried in the incubator until reaching a constant weight, and later kept in the freezer. To determine the optimum fermentation time, three different production cycles were carried out to analyze lactic acid bacteria, enzyme activity and soluble protein. To count lactic acid bacteria, 1 g samples of SMFL were serially diluted (1:10) in 0.65% sterile saline solution and plated on MRS (Man Rogosa Sharpe) agar culture medium (Kasvi, São José dos Pinhais. Brazil). Fermented soybean samples plated in Petri dishes were placed in an incubator, and the temperature was maintained at 35 °C. Total counts of colony-forming units (CFU) were performed after 48 h of incubation on agar.

Analyses of the enzymatic activity of α-amylase and total alkaline proteases were performed at different times of SMFL fermentation. Samples were placed in 50 mL tubes where they were diluted in chilled distilled water (1:6, *w*/*v*). The tubes were centrifuged (centrifuge K14-4000, Kasvi, São José dos Pinhais, Brazil) for 15 min at 4000 rpm, and the supernatants were used in the analyses. All assays were analyzed in duplicate in the three production runs described above. Alpha-amylase activity was measured (spectrophotometer Spectroquant Pharo300, Merck, Darmstadt, Germany) at λ = 580 nm using soluble starch (0.3%) as substrate in Na_2_HPO_4_ buffer solution (pH 7.4) [40]. A unit of α-amylase activity (U) was defined as a milligram of starch hydrolyzed in 30 min at 25 °C per milliliter of the supernatant.

The activity of total alkaline proteases was determined after 30 min of incubation (dry block incubator Model K80-200, Kasvi, São José dos Pinhais, Brazil) at 25 °C, using 0.5% (*w*/*v*) casein as substrate in 50 mM Tris-HCl (pH 8.0). The reaction was stopped with trichloroacetic acid (20% *w*/*v*), the mixture was centrifuged (centrifuge K14-4000, Kasvi, São José dos Pinhais, Brazil) (5000 rpm, 20 min) and the absorbance of the supernatant was measured (spectrophotometer Spectroquant Pharo300, Merck, Darmstadt, Germany) at λ = 280 nm at room temperature. A unit of total alkaline protease activity (U) per milliliter was defined as 1 µmol of casein hydrolyzed per minute per milliliter of supernatant [41]. In order to measure the soluble protein, the fermented soybean meal samples were macerated until obtaining a fine powder. Then, 50 mM Tris-HCl buffer, pH 7.0, containing protease inhibitors (1 mM phenylmethylsulfonyl fluoride, 1 mM benzamidine, 2 mM thiourea and 10 mM EDTA) and 2% (*m*/*v*) polyvinylpolypyrrolidone, was added. Soluble protein was measured according to the Bradford method [42], using bovine serum albumin (Sigma-Aldrich, St. Louis, MO, USA) as a standard.

Based on lactic acid bacteria counts, enzymatic activity and bromatological composition, the time of 48 h for SMFL fermentation was determined. Samples were collected before and after fermentation for bromatological analysis [43], pH and amino acid profile [44] (Table 1). The amino acid composition of SMFL was determined according to White, Harty and Fry [44]. The pH was measured in solution, according to the potentiometric method described by the Association of Official Analytical Chemists [45]. Ten grams of the samples were placed in a 250 mL Erlenmeyer flask, where 100 mL of distilled water at 25 °C was added. After adding water, the contents were mixed in an electronic shaker (model DT3110H, DiagTech, São Paulo, Brazil) for 30 min. The contents were placed in a 500 mL beaker and allowed to rest for 10 min, and then pH (model AT-355, Alfakit, Florianópolis, Brazil) was measured.

### 2.3. Experimental Diets

Five isoproteic (39% crude protein) and isoenergetic (about 4300 kcal of gross energy kg^−1^) diets were formulated, according to the nutritional requirement of the South American catfish [47], with five levels of inclusion (0%, 7%, 14%, 21% and 28%) of SMFL (Table 2). Dietary levels of crude lipids were 11.1%, 10.4%, 9.7%, 7.1% and 7.9%. The adjustment in the lipid levels was necessary to balance the energy levels of the diets. In addition to the fermented soybean meal, diets were formulated using corn and soybean oil as energy sources and fishmeal and soybean meal as protein sources. Marine fishmeal was purchased from Agroforte (Laguna, Santa Catarina, Brazil). The ingredients were crushed in a knife mill with a 2 mm diameter mesh sieve, mixed, pelleted (meat grinder MTU 08, Arbel, São José do Rio Preto, Brazil) with the addition of water (30%) and placed in an incubator (SSDcr-336L, SolidSteel, Piracicaba, Brazil) at 45 °C for 36 h. The proximal composition of experimental diets was analyzed using the methods described in AOAC [43]. The diets were stored in plastic containers and kept under refrigeration (4 °C) until the moment of use.

### 2.4. Animals and Facilities

Fish were acquired from Piscicultura Nossa Senhora Aparecida, located in the city of Ijuí, Rio Grande do Sul (Brazil), and previously acclimated to the experimental conditions. A total of 240 South American catfish juveniles (3.0 ± 0.5 g) were used, which were distributed in 20 polyethylene tanks with a useful volume of 70 L. The tanks were connected to a water recirculation system equipped with a mechanical and biological filter and a heating system. Fish were fed until apparent satiety, once a day, at 9 a.m. This feeding protocol has already been determined as suitable for the South American catfish [48].

Every day, organic waste was removed from the bottom of the tanks, and the presence of dead animals in the tanks was checked. Temperature (28.5 ± 0.3 °C), pH (7.12 ± 0.25) (HI98130, Hanna, Brazil) and dissolved oxygen (6.41 ± 0.54 mg L^−1^) (HI9147-10, Hanna, Barueri, Brazil) were monitored daily. Total ammonia was checked weekly (LabconTest, Alcon Pet, São Francisco de Assis, Brazil) (0.17 ± 0.2 mg NH_3_ L^−1^). Salinity was maintained around 4 g L^−1^ [49] in all polyethylene boxes throughout the experimental period. Water quality parameters remained within the recommended parameters for the cultivation of the South American catfish [50].

### 2.5. Productive Performance and Sample Collection

At the beginning and at the end of the experiment, biometrics were performed to assess the animals’ growth. All fish were anesthetized in a clove oil solution (1 g 10 L^−1^ of water) and individually weighed on a semianalytical digital scale with a precision of 0.01 g (model ML 600, Marte, São Paulo, Brazil). The following zootechnical performance indicators were evaluated: final weight, specific growth rate (SGR), apparent feed conversion (FC), feed ingestion (FI) and weight gain (WG). Mortality was recorded to assess the survival rate (S).


(1)
$$\mathrm{SGR}\left(\%.{\mathrm{day}}^{-1}\right)=\left(\frac{\mathrm{In}\;\mathrm{final}\;\mathrm{weight}-\mathrm{In}\;\mathrm{initial}\;\mathrm{weight}}{\mathrm{experimental}\;\mathrm{period}}\right)\times 100$$



(2)
$$\mathrm{FC}=\frac{\mathrm{feed}\;\mathrm{consumption}}{\mathrm{total}\;\mathrm{weight}\;\mathrm{gain}}$$



(3)
$$\mathrm{FI}\left(g\right)=\frac{\mathrm{total}\;\mathrm{feed}\;\mathrm{consumption}}{\mathrm{tank}}$$



(4)
$$\mathrm{WG}\left(g\right)=\mathrm{final}\;\mathrm{average}\;\mathrm{weight}\;\left(\mathrm{FW}\right)-\mathrm{initial}\;\mathrm{average}\;\mathrm{weight}\;\left(\mathrm{IW}\right)$$



(5)
$$S\left(\%\right)=\left(\frac{\mathrm{total}\;\mathrm{animals}\;\mathrm{harvested}}{\mathrm{total}\;\mathrm{animals}\;\mathrm{stocked}}\right)\times 100$$


At the end of the experiment, two fish per replicate (eight fish per dietary treatment) were collected, anesthetized and euthanized for analysis of intestinal histomorphometry, intestinal microbiology and gastrointestinal tract enzyme activity.

### 2.6. Count of Total Heterotrophic, Lactic Acid and Vibrionaceae Bacteria

The intestinal tracts were homogenized, serially diluted (1:10) in sterile saline solution at 0.65% and seeded in the culture media with MRS (Man Rogosa Sharpe) agar (Kasvi, Brazil), TSA (tryptone soy agar) (Kasvi, São José dos Pinhais, Brazil) and TCBS (thiosulfate, citrate, bile and sucrose) agar (Acumedia, Indaiatuba, Brazil) to quantify lactic acid, total heterotrophic and *Vibrionaceae* bacteria, respectively. Intestinal samples were plated on Petri dishes and placed in incubators at 36 °C. Total counts of colony-forming units (CFU) were performed after 24 h of incubation in TSA and TCBS agar. In the MRS medium, the counts were performed after 48 h of incubation according to Jatobá et al. [51].

### 2.7. Enzyme Analysis

Stomachs and intestines were extracted through a longitudinal incision in the abdominal cavity and immediately frozen at −80 °C until the moment of analysis. The gastrointestinal tracts were weighed, ground and added in 50 mL tubes. The homogenates and total alkaline protease activity analysis were realized as described in Section 2.2. Lipase activity was measured at λ = 410 nm using *p*-nitrophenyl laurate (3 mM) in propanol as substrate [52]. The reaction was stopped by the addition of acetone [53]. A unit of lipase activity (U) was defined as the amount of enzyme required for the hydrolysis of 1 µmol of *p-*nitrophenyl laurate in 20 min at 25 °C per milliliter of enzyme extract.

### 2.8. Intestinal Morphometry

Sections of 3 cm in length were collected from the midgut (4 cm after the junction of the stomach and the intestine) and dipped in a 10% buffered formalin fixative solution for 24 h, which was replaced by 70% alcohol, remaining preserved until the evaluation of the morphohistological characteristics of the mucosa. For this purpose, the intestinal sections were cut into slices of approximately 0.3 cm, dehydrated with graded series of ethanol and embedded in histological paraffin. Paraffin blocks were cut in a rotating microtome, with 5 µm thick cuts.

Two slides were made from each sample, and the sections were stained according to the Harris eosin staining technique [54]. Slides were photographed using a digital camera (300 dpi) coupled to a microscope (10× objective). Fifteen villi of each fish were measured. Total height and height (corresponding to the distance from the apex of the villi to the beginning of the muscle layer and from the apex of the villi to the end of the serosa, respectively) and villous width values were measured using the ToupTek (Hangzhou, China) ToupView-x64 image analyzer software, version 2270/07/03.

### 2.9. Statistical Analysis

All data were subjected to tests to verify the normality of errors (Shapiro–Wilk test) and homoscedasticity of variances (Levene test). Broken-line model, linear, and polynomial regressions were performed using the SAS statistical program, version 9.0. The model with the best coefficient of determination was chosen to estimate the response.

## 3. Results

The lactic bacteria count (Figure 1A; *p* = 0.0002; R^2^ = 0.86) and amylase activity (Figure 1B; *p* = 0.0106; R^2^ = 0.56) during soybean meal fermentation showed better fit to a quadratic equation. Values increased faster in the first 48 h. Protease activity (Figure 1C; *p* < 0.0001; R^2^ = 0.67) and soluble protein (Figure 1D; *p* < 0.0001; R^2^ = 0.85) showed a linear response with their values as fermentation time increased.

No fish mortality was observed throughout the experiment. Regarding growth and feed performance variables, the broken-line analysis showed that dietary SMFL levels close to 21% may be included in the diets for South American catfish juveniles without any negative effect on somatic weight gain (Figure 2A; *p* = 0.0002), specific growth rate (Figure 2B; *p* = 0.0002) or apparent feed conversion (Figure 2C; *p* = 0.0178) of South American catfish juveniles. There was a trend (mean 31.21 ± 3.84, *p* = 0.0978) towards a reduction in feed ingestion in inclusions of SMFL higher than 14%.

The count of *Vibrionaceae* in the intestine of South American catfish juveniles fed with SMFL showed a quadratic effect (Figure 3; *p* = 0.0316). Among the evaluated treatments, the lowest counts were observed in fish fed the diets containing 7%, 14% and 21% SMFL. There was no effect of SMFL inclusion on the count of heterotrophic bacteria (mean 3.32 ± 0.93 CFU g^−1^, *p* = 0.9827) or lactic acid bacteria (mean 3.01 ± 0.93 CFU g^−1^
*p* = 0.30)

The inclusion of SMFL did not modify the specific activity of total alkaline proteases (mean 1.43 ± 0.87 U mg^−1^ protein, *p* = 0.5404) or lipase (mean 1.47 ± 079 U mg^−1^ protein, *p* = 0.4612) in the digestive tract of South American catfish juveniles. Intestinal histomorphometry was also not affected. The inclusion of SMFL did not change the total height (mean 616.23 ± 144.94 µm, *p* = 0.3482), height (mean 481 ± 121.73 µm, *p* = 0.7533) or width (mean 88.09 ± 14.33 µm, *p* = 0.3486) of the intestinal villi of South American catfish juveniles.

## 4. Discussion

During fermentation, probiotic bacteria multiply and produce metabolites with bioactive properties [20]. Lactic acid bacteria increased in the SMFL from 0.00 to 5.80 CFU g^−1^ after 48 h of fermentation. These values are in agreement with the literature and demonstrate that the colonies grew as expected [18,55]. The colonies of lactic acid bacteria grew exponentially in the first 24 h of fermentation. After 48 h, growth stabilized, and this was considered the optimal fermentation time. As nutrients decrease and metabolites accumulate, the culture enters a stationary phase and reduces the production of organic compounds of interest [56]. As a result of fermentation, an increase in soluble protein and amino acids was observed. The increase in soluble protein is related to the presence of low-molecular-weight bioactive peptides [57] These peptides are also found in protein hydrolysates and can improve fish growth and intestinal health [58]. The synthesis of amino acids may be the result of lactic acid fermentation [8] and indicates an improvement in the nutritional value of SMFL. There was an increase in the activity of protease and amylase enzymes in SMFL. During fermentation, different types of enzymes are secreted by *Lactobacillus* [18], and SMFL can be a source of exogenous enzymes. Lactic acid is also secreted during fermentation, which leads to a drop in pH and can benefit the digestive process [59]. In the present study, the pH values remained relatively stable; the reduction in pH was from 6.64 to 6.47. It was not a significant decrease, but similar results were obtained using another bacterial species, *Bifidobacterium animalis*, in soybean fermentation [55].

The dietary inclusion of up to 21% SMFL replacing fishmeal did not affect the zootechnical performance of the South American catfish juveniles. The zootechnical parameters are within what is expected for the species [58,60], and this result confirms the efficiency of fermentation as a strategy to include a higher percentage of soybean meal in carnivorous fish feeds. It has been shown that soybean meal fermentation reduced the amounts of antinutritional factors and allowed greater inclusion of soybean meal in the largemouth bass (*Micropterus salmoides*) diets [25]. Previous studies showed that an inclusion of above 35% soybean meal worsened the growth of the South American catfish [37]. In the present study, it was possible to include up to 46% soybean meal, considering the ingredients in the fermented and in the unfermented form. The higher inclusion of soybean meal may have been made possible by the improvement in the nutritional characteristics of the SMFL. Soybean meal fermentation increased soluble protein, indicating a higher proportion of low-molecular-weight peptides [57]. These peptides find specific absorption sites in the intestine, which results in more efficient absorption and thus compensates for the lower inclusion of fishmeal in diets [61].

The highest level of SMFL inclusion (28%) reduced the weight gain of South American catfish juveniles. The result obtained in this experiment corroborated other studies that included levels of fermented soybean meal between 24% and 30% [3,62,63,64]. At the highest levels of inclusion, there was a trend towards a reduction in food intake, indicating that possibly the antinutritional factors were not completely removed. In addition, another possible explanation is the increase in the soluble protein present in the fermented meal. At high levels of inclusion, excessive peptides can compromise their absorption due to saturation of transport mechanisms and thus reduce animal performance [65,66,67,68]. Furthermore, there may be an increase in amino acid oxidation and a reduction in dietary protein retention due to the large number of free peptides present in the fermented meal used in this study, which may also compromise their absorption [69]. These results indicated that more studies are still needed to clarify how the use of fermented ingredients affects protein absorption and how this knowledge may be applied for guaranteeing higher levels of SMFL in aquafeeds.

Increasing the inclusion of SMFL in the diets of South American catfish led to a reduction in dietary crude lipids. This adjustment was necessary to balance the energy levels of the diets. Although it is a protein source, SMFL does have moderate levels of starch and other carbohydrates that can be used as an energy source. South American catfish does not use carbohydrates as well as other omnivorous freshwater species, but it can adapt to different carbohydrate sources in diets [70]. Furthermore, if the energy intake had not been sufficient, an increase in the ingestion of these diets would be expected to compensate [71]. The result was the opposite: there was a tendency to reduce feed ingestion with the increase in the inclusion of SMFL in diets. Furthermore, we also find no differences in growth performance among fish fed diets differing in their crude lipid content (7.1–11%), which indicated that nutritional lipid requirements were already covered with the lowest crude lipid content tested (21% SMFL diet). These results are of relevance since they indicate that it is possible to reduce dietary lipid inclusion levels in diets for South American catfish juveniles when dietary crude protein (37.5–39.2%) and energy levels (4205–4300 kcal g^−1^) are covered.

In terms of microbial load in the intestinal mucosa, the inclusion of 21% SMFL promoted a decrease in bacteria from the family *Vibrionaceae* in the intestine of the South American catfish when compared to the control group. Cheng et al. [72] obtained a similar result with shrimp (*Litopenaeus vannamei*) when using fermented soybean meal, causing the reduction of two species of *Vibrionaceae* bacteria. Other authors have demonstrated that fermented soybean meal can improve the composition of the intestinal microbiota in fish [23,25,26]. Most of the bacteria in the fermented products die during the drying process; however, the residues of dead bacteria and their metabolites produced may have contributed to these positive results [25]. The fermented meal would work as a paraprobiotic additive in which the microorganisms are not in their viable form but still bring benefits to intestinal health [73]. Another explanation for the decrease in *Vibrionaceae* bacteria would be the presence of bioactive peptides that have antimicrobial activities and that are abundantly present in soybean meal [74]. It should be noted that at the highest level of SMFL inclusion, the same response was not obtained in terms of controlling *Vibrionaceae* bacteria. As performance was compromised in this treatment, the bacterial count may also have been affected.

Changes in the bacterial counts of *Vibrionaceae* were not related to changes in the abundance of heterotrophic and lactic acid bacteria in the intestine of South American catfish, which suggested that these changes in the *Vibrionaceae* were not due to competitive exclusion mechanisms with heterotrophic and lactic acid bacteria. Fermented ingredients are sources of probiotic bacteria and contain functional metabolites that can positively affect the intestinal microbiota of fish [73]. In the present study, due to technical limitations, only two fish were collected per replicate for each analysis. The low number of repetitions may have prevented the detection of differences between treatments. More studies are needed to understand the effects of ingesting fermented ingredients on intestinal bacteria counts. It is also necessary to evaluate, through molecular biology studies using massive sequencing approaches, the dietary effects of SMFL on autochthonous microbiota.

The enzymatic activity and the intestinal morphometry of South American catfish juveniles were not affected by the inclusion of SMFL. The use of fermented ingredients can bring benefits to intestinal health, such as increased enzymatic activity [27,28,29] and improvement of intestinal villi [16,27,31,32,33,34]. However, these beneficial effects were not seen in this study. One of the potential reasons for explaining such results was the low number of replicates used, which may have hampered detecting statistical differences among dietary groups. On the other hand, there are several factors involved in the fermentation process that may affect the functional properties of the fermented product [8]. Ranjan et al. [75] and Azarm and Lee [39] also did not observe changes in the digestive enzymes of *Labeo rohita* and *Acanthopagrus schlegeli* fed with fermented soybean meal. Even with no changes, the obtained results can be considered positive, as high inclusions of soybean meal can compromise the enzymatic activity [76,77] and the intestinal morphometry of fish [6]. Previous studies have shown that soybean meal fermentation can protect the fish intestinal epithelium from possible damage [25,78].

## 5. Conclusions

Fermentation of soybean meal with *L. acidophilus* improved its nutritional profile and microbiological load. In particular, there was an increase in lactic acid bacteria count, enzymatic activity and soluble protein in the SMFL. The inclusion of SMFL up to 21% as a strategy for replacing fishmeal did not affect zootechnical performance in terms of growth and feed efficiency performances, nor did it affect intestinal morphology or enzymatic activity (total alkaline proteases and lipase) in the gastrointestinal tract of South American catfish juveniles. The amount of the Gram-negative bacteria from the *Vibrionaceae* family in the intestine decreased with the inclusion of up to 21% SMFL. These results demonstrate that soybean meal fermentation with *L. acidophilus* enabled a greater inclusion of soybean protein in diets and also promoted the control of intestinal pathogenic bacteria.

## Figures and Tables

**Figure 1 animals-12-00690-f001:**
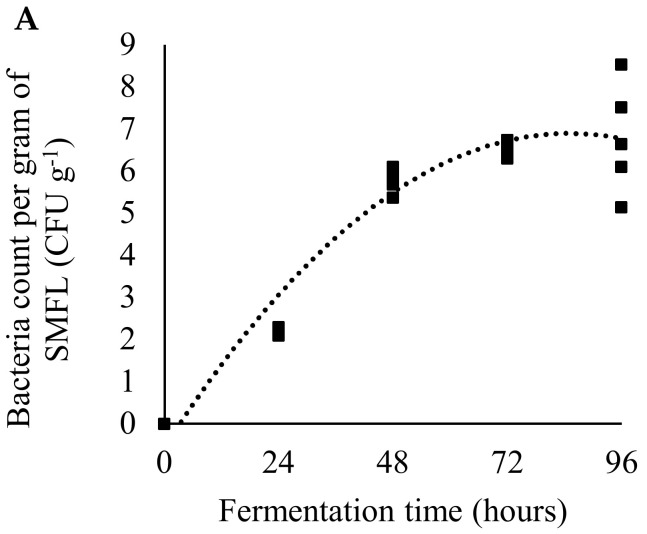
Lactic acid bacteria count (**A**), amylase activity (**B**), protease activity (**C**) and soluble protein (**D**) of soybean meal fermented with *Lactobacillus acidophilus* (SMFL) at different times of fermentation. Lactic acid bacteria count (CFU g^−1^): y = −0.001x^2^ + 0.1752x − 0.5475, R^2^ = 0.86, *p* = 0.002. Amylase activity (U g^−1^): y = −0.000041x^2^ + 0.0062x − 0.0438, R^2^ = 0.56, *p* = 0.0106. Protease activity (U g^−1^): y = 0.000075 + 0.00736, R^2^ = 0.67, *p* < 0.0001. Soluble protein (mg g^−1^): y = 0.2830x + 22.6624, R^2^ = 0.85 *p* < 0.0001.

**Figure 2 animals-12-00690-f002:**
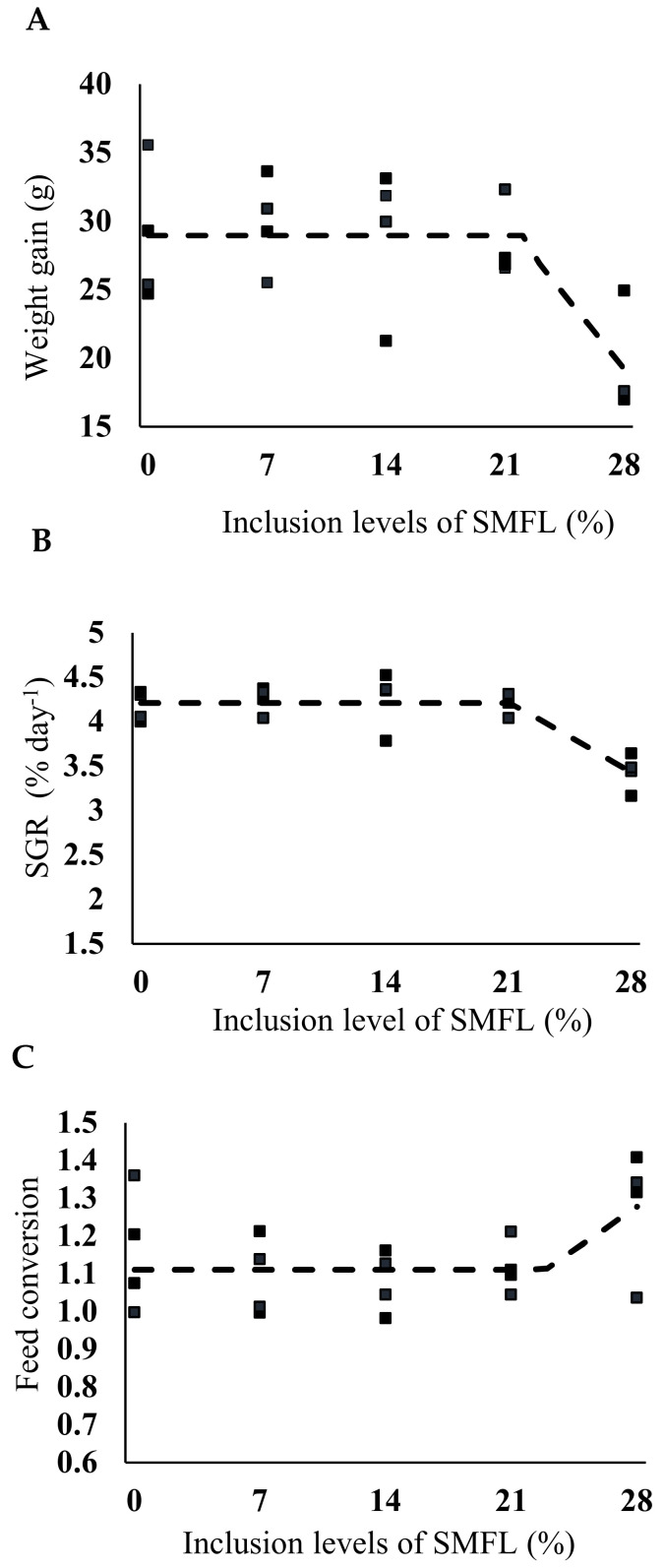
Weight gain (**A**), specific growth rate (SGR) (**B**) and feed conversion (**C**) of South American catfish fed with diets containing different levels of soybean meal fermented by *Lactobacillus acidophilus* (SMFL). Weight gain: Y = 28.9506 − 1.5489(X − 21.6785), Y = 28.9506 when X < 21.6785. The breakpoint of the broken line is 21.6% (*p* = 0.0002). SGR: Y = 4.2187 − 0.1134(X − 21.0458), Y = 4.2187 when X < 21.0458. The breakpoint of the broken line is 21.04% (*p* < 0.0001). Feed conversion: Y = 1.1106 + 0.0326(X − 22.8846), Y = 1.1106 when X < 22.8846. The breakpoint of the broken line is 22.8% (*p* = 0.0178).

**Figure 3 animals-12-00690-f003:**
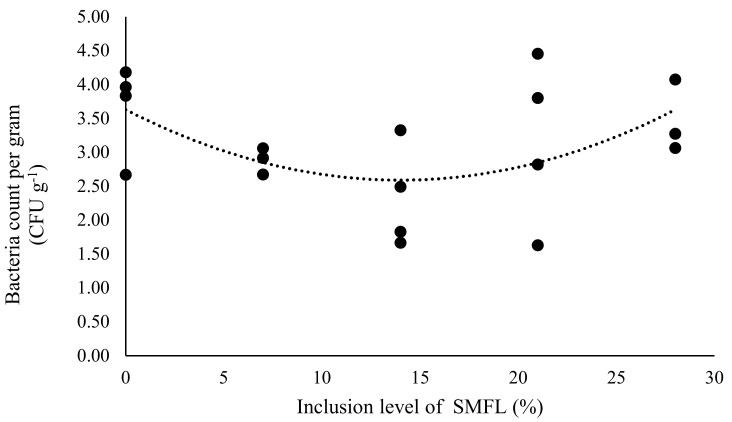
Count of *Vibrionaceae* (CFU g^−1^) in the intestine of South American catfish fed with diets containing different levels of soybean meal fermented by *Lactobacillus acidophilus*. Y = 0.0053x^2^ − 0.1484 + 3.6279, R^2^ = 0.2755, *p* = 0.0316.

**Table 1 animals-12-00690-t001:** Analysis of bromatological composition, pH and amino acids of soybean meal and soybean meal fermented by *Lactobacillus acidophilus* (SMFL).

	Soybean Meal *	SMFL
*Proximal composition (%)*		
Crude protein	44.82	43.99
Ethereal extract	5.23	4.19
Dry matter	88.35	90.64
Mineral matter	7.42	7.75
pH	6.64	6.47
*Essential amino acids (%)*		
Arginine	3.35	3.47
Histidine	1.20	1.31
Isoleucine	2.13	2.00
Leucine	3.51	3.84
Lysine	2.87	3.14
Methionine	0.61	0.60
Phenylalanine	2.34	2.73
Valine	2.22	2.38
Threonine	1.78	1.98
*Nonessential amino acids (%)*		
Proline	2.31	2.63
Cystine	0.67	0.72
Alanine	2.03	2.33
Aspartic acid	3.22	5.86
Glutamic acid	4.44	9.12
Glycine	1.97	2.15
Serine	2.46	2.70
Tyrosine	1.66	1.74
Sum of amino acids (%)	38.77	48.70

* Amino acid profile according to Rostagno [46].

**Table 2 animals-12-00690-t002:** Formulation and analyzed composition of experimental diets.

	Experimental Diets				
Ingredients (%)	SMFL 0%	SMFL 7%	SMFL 14%	SMFL 21%	SMFL 28%
Soybean meal	25.0	25.0	25.0	25.0	25.0
Corn	9.5	9.0	8.5	8.5	8.5
Wheat flour	12.0	12.0	12.0	12.0	12.0
Fishmeal	46.5	40.5	34.5	28.0	22.0
SMFL	0.0	7.0	14.0	21.0	28.0
Soybean oil	6.0	5.5	5.0	4.5	3.5
Premix ^1^	1.0	1.0	1.0	1.0	1.0
Centesimal composition					
Crude protein (%)	39.19	38.74	38.43	37.93	37.45
Gross energy (kcal kg^−1^)	4298.93	4314.77	4205.53	4219.67	4333.41
Ethereal extract (%)	11.05	10.38	9.71	7.09	7.87
Mineral matter (%)	14.62	12.86	11.57	10.27	9.63
Dry matter (%)	91.21	91.38	91.50	91.79	92.30
Essential amino acids (%)					
Lysine	3.38	3.22	3.08	2.93	2.88
Methionine	1.00	0.92	0.85	0.77	0.70
Arginine	2.18	2.26	2.36	2.44	2.48
Threonine	1.83	1.82	1.77	1.73	1.71
Leucine	3.51	3.45	3.41	3.36	3.32
Isoleucine	2.46	2.37	2.28	2.19	2.12
Valine	2.51	2.43	2.36	2.27	2.22
Histidine	1.08	1.07	1.06	1.05	1.05

^1^ Folic acid—2400 mg, nicotinic acid—48 g, pantothenic acid—24 g, biotin—96 mg, vit. A—2,400,000 IU, vit. D3—400,000 IU, vit. E—24,000 IU, vit. B1—9600 mg, vit. B2—9600 mg, vit. B6—9600 mg, vit. B12—9600 mg, vit. K3—4800 mg, vit. C—96 g, iron—100 g, manganese—40 g, zinc—6000 mg, cobalt—20 mg, iodine—200 mg, selenium—200 mg, antioxidant—19.6 g. Soybean meal fermented by *Lactobacillus acidophilus* (SMFL).

## Data Availability

Data can be available upon direct request to the corresponding author.

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
