# Peer review of "Fermentation of Soybean Meal with Lactobacillus acidophilus Allows Greater Inclusion of Vegetable Protein in the Diet and Can Reduce Vibrionacea in the Intestine of the South American Catfish (Rhamdia quelen)"

_animals, 2022, doi:10.3390/ani12060690_

Round 1
Reviewer 1 Report
This is a well-executed study that embraces the classical evaluation of a novel processed product namely a fermented soybean subjected to lactobacillus with a view to testing as an alternative protein source for the S American catfish. This species is of importance in Brazil where the trials were undertaken. From this angle it is a novel investigation although very much in the style of many other publications for different species. The idea of fermenting plant ingredients such as soybean is itself not too novel but certainly relevant and new for the catfish. In the study, the standard soyabean inclusion in all diets was fixed at 25% and the SMFL increased by substitution of the fishmeal itself. This is OK as it shows how far we can reduce fishmeal further. It would have been interesting to have possibly reduced the standard soy first to see if there were any advantages.
The introduction could include some more recent work linked to this topic on other species like bass, bream and tilapia. try to insert some related work on say DDGS Distillers grain and solubles etc ??? one or two 2020/21 citations would be good!
In effect there was little difference in growth performance metrics and intestinal enzyme activities. There was no real evidence of any effects in terms of protease, amylase reductions. A corresponding treatment to compare the fermented with standard soya at the highest fishmeal substitution could have shown differences attributable to the levels of ANF’s in the soybean meal ingredient. In this sense, experimental design was constrained and could have been better. The grammar and English could be minor refined in places and tenses should be checked but are mainly correct. The figures and legends in the graphics need better description to emphasize that you are measuring enzyme activities and lactic acid bacteria in the gut/intestine of fish and not in the fermented product or other way round??? Figure 1 caption below is rather badly stated and you need to work on this. restructure please !!!
This caption should be restated as it makes no sense as written. very confusing indeed! please revisit ALL your figure captions. Also can you add error bars to the data presented ? this would give more confidence to the presentation of the data for each level of fermented soybean inclusion and effects.
Figure 1. may be included in the diets for South American catfish juveniles without any prejudice 262 on somatic weight gain (Figure 2A; P = 0.0002), specific growth rate (Figure 2B; P = 0.0002) and 263 apparent feed conversion (Figure 2C; P = 0.0178) of South American catfish juveniles. There was a 264 trend (mean 31.21±3.84 P = 0.0978) towards a reduction in feed ingestion in inclusions higher than 265 14% of SMFL
There are interesting findings here that are worthy of being disseminated but they are not ground breaking but add to our knowledge base for effective feed formulations. It should be possible to mention the fact in the discussion concerning how the fermented soy could influence the gut microbiome as well as intestinal lactobacillus flora. Please refer and modest corrections are needed to improve the images and descriptions.
Reviewer 2 Report
General comments
This manuscript deal is the study the effect of diets containing different inclusion levels of soybean meal fermented by Lactobacillus acidophilus, and if those effects justify the use of this soybean meal pre-treatment.
Soy bean fermentation is not an original methodology for its inclusion y fish diets. There are several studies that afford this methodology in different fish species cultures. But its use in American Catfish is the first time reported.
The experimental procedure has several gaps. The most important is the number of fish samples sampled. In my opinion, just two animals per tank is not enough, first to do a complete statistical analysis and second to obtain accurate results as the authors mention in the "Discussion" section. Also, they make only technical duplicates of every assay (enzyme activity, morphometrical, etc.) and this is not adequate for a complete statistical analysis.
Title is a bit longer. Try to reduce the number of words. “Fermentation of soybean meal allows greater inclusion and improve intestine health of south American catfish …”
Introduction section should be rearranged. For example, paragraph from lines 72 to 81 could be before paragraph from lines 62 to 71, because in my opinion introducing the fermentation process should be before the explanation of soybean fermentation.
Also, I think that the only property of Lactobacillus that make them useful for its use as probiotics is the production of antimicrobial compounds. Maybe, paragraph from lines 72 to 81 should be rewritten, taken into consideration this. I encourage authors to find more properties of Lactobacillus (or lactic acid bacteria) that make then attractive to be used as probiotics.
The wording of the section “Materials and methods” should be improved.
For example, the same methodologies are repeated in several sections (e.g., in 2.2. and 2.7). In my opinion, only writing the method once is enough and when it is mentioned again just describe in detail how the samples are processed.
The explanations of these methods should also be improved by adding more details and including the brands of the products and equipment used.
Regarding statistics, it should be explained in more detail how it has been carried out, as well as the programs used, for the study of the normality of the errors and the homoscedasticity of the variances.
In this section, in lines 143 from 153 is explained why a fermentation of 48 hours is chosen. In my opinion, this explanation must be in Results section, as well as, the results obtained in the assays using and specifying the times experimented for the other fermentations (bacterial counts, enzymatic activities and protein concentrations). Without this data, there is no evidences of why those 48 hours is the best time for formation of soybean meal.
In the Results section and its figures, several errors in the wording must be corrected, as well as figure 1 caption. Also, results from zootechnical performance indicators are not included in the text, only there is a figure.
Regarding Discussion section, I would like to question why authors explain, in paragraph from lines 348 to 360, resume several studies that explain why family Vibrionaceae bacteria decrease when using fermented soybean meal, mainly due to the presence of residues and metabolites from bacteria used in fermentation. But how this explanation could give us and explanation about this decrease in the intestine of that fish species, when this reduction follows a polynomial regression curve? And, if the initial number of lactic acid bacteria in fermentation process are the same in all diets?
Finally, in last sentences of paragraph from lines 366 to 372, it is stated that more samples are needed to detect differences between treatments. So, in my opinion this is a missense of the tittle affirmation of “reduces pathogenic bacteria in the intestine”. I agree with authors that more samples were needed in this study.
Specific comments
- Please specify the strain of Lactobacillus acidophilus used in this study.
- In line 108 is not clear if the total number cfu of acidophilus used is the total added. Also, lines 109 and 110 are the same. Please rewrite paragraph from lines 104 to line 114, for a better understand of the process.
- The brand of the Culture medium used must be indicated. As well as the brands of the reactive and instruments used for the other analysis.
- In line 125, authors use the commercial brand “Falcon”. Instead of this 50 ml, or whatever capacity, tube must be used.
- In the intestinal morphometry methodology, it not mentioned the inclusion material microtome used.
- In figure 1 it is not respecified how many hours has been the fermentation process. As I mentioned before, this table should include the same results for the other fermentation times.
- The caption of figure 1 is from figure 2.
Reviewer 3 Report
Analytical method of amino acid composition of SMFL and diet should be shown.
Result of feeding trial is better to be shown as Table, show result of statistical analysis. Feed ingestion rate that author mentioned seems to be slightly different between result and discussion (14% in result, highest level in discussion??). And author discussed protein retention, but no data was shown. These data are important for understanding of result.
L285-289;Digestive enzyme activity and morphological change were described average of all treatments only. Show values of each treatment. Table or figure is better.
L262-266; Revise.
Changes of amylase activity of intestine is not found, although analytical method of amylase of intestine is shown in M&M.
Table 1; Sum of amino acids exceeds crude protein content. Why? Sum of amino acids of SMFL is 9% higher than SBM. Fermentation of SBM increased amino acid content?
Tryptophan is not appeared. Alanine appears in both EAA and NEAA.
Check data and calculation.
Table 3; Phenylalanine and tryptophan, which are EAA of fish, are not shown.
Fig 1; Legend of Figure 2 is missing
Fig 2; Check SGR. Data and unit are wrong.
Round 2
Reviewer 2 Report
I still recommend reduce the number of words in title, and not include the affirmation that "reduce pathogenic bacteria in the intestine".
Reviewer 3 Report
I have no comment on revised version.
Author Response
Thank you for the corrections